# Mitochondrial DNA Copy Number Adaptation as a Biological Response Derived from an Earthquake at Intrauterine Stage

**DOI:** 10.3390/ijerph182211771

**Published:** 2021-11-10

**Authors:** Jonatan A. Mendoza-Ortega, Enrique Reyes-Muñoz, Sonia Nava-Salazar, Sandra Rodríguez-Martínez, Sandra B. Parra-Hernández, Lourdes Schnaas, Blanca Vianey Suárez-Rico, Libni A. Torres-Olascoaga, Andrea A. Baccarelli, Rosalind J. Wright, Robert O. Wright, Guadalupe Estrada-Gutierrez, Marcela Tamayo-Ortiz

**Affiliations:** 1Immunobiochemistry Department, Instituto Nacional de Perinatología Isidro Espinosa de los Reyes, Montes Urales 800, Lomas de Virreyes, Mexico City 11000, Mexico; jonatan.mdz93@gmail.com (J.A.M.-O.); s_nava_s@yahoo.com.mx (S.N.-S.); rebe1602@hotmail.com (S.B.P.-H.); 2Innate Immunology Laboratory, Immunology Department, Escuela Nacional de Ciencias Biológicas-Instituto Politécnico Nacional, Mexico City 11350, Mexico; sandrarodm@yahoo.com.mx; 3Coordination of Gynecological and Perinatal Endocrinology, Instituto Nacional de Perinatología Isidro Espinosa de los Reyes, Ministry of Health, Mexico City 11000, Mexico; dr.enriquereyes@gmail.com; 4National Institute of Perinatology, Mexico City 11000, Mexico; lschnaas@hotmail.com; 5Research Branch, Instituto Nacional de Perinatología Isidro Espinosa de los Reyes, Montes Urales 800, Lomas de Virreyes, Mexico City 11000, Mexico; blancasuarezrico@gmail.com (B.V.S.-R.); gpestrad@gmail.com (G.E.-G.); 6National Institute of Public Health, Cuernavaca 62100, Mexico; libniavib@gmail.com; 7Department of Environmental Health Sciences, Columbia University, New York, NY 10029, USA; andrea.baccarelli@columbia.edu; 8Department of Pediatrics, Icahn School of Medicine at Mount Sinai, New York, NY 10029, USA; rosalind.wright@mssm.edu (R.J.W.); robert.wright@mssm.edu (R.O.W.); 9Department of Environmental Medicine and Public Health, Icahn School of Medicine, New York, NY 10029, USA; 10Research Unit in Occupational Health, Mexican Social Security Institute, Mexico City 06720, Mexico

**Keywords:** mtDNAcn, prenatal, DOHAD, earthquake

## Abstract

An altered mitochondrial DNA copy number (mtDNAcn) at birth can be a marker of increased disease susceptibility later in life. Gestational exposure to acute stress, such as that derived from the earthquake experienced on 19 September 2017 in Mexico City, could be associated with changes in mtDNAcn at birth. Our study used data from the OBESO (Biochemical and Epigenetic Origins of Overweight and Obesity) perinatal cohort in Mexico City. We compared the mtDNAcn in the umbilical cord blood of 22 infants born before the earthquake, 24 infants whose mothers were pregnant at the time of the earthquake (exposed), and 37 who were conceived after the earthquake (post-earthquake). We quantified mtDNAcn by quantitative real-time polymerase chain reaction normalized with a nuclear gene. We used a linear model adjusted by maternal age, body mass index, socioeconomic status, perceived stress, and pregnancy comorbidities. Compared to non-exposed newborns (mean ± SD mtDNAcn: 0.740 ± 0.161), exposed and post-earthquake newborns (mtDNAcn: 0.899 ± 0.156 and 0.995 ± 0.169, respectively) had increased mtDNAcn, *p* = 0.001. The findings of this study point at mtDNAcn as a potential biological marker of acute stress and suggest that experiencing an earthquake during pregnancy or before gestation can have programing effects in the unborn child. Long-term follow-up of newborns to women who experience stress prenatally, particularly that derived from a natural disaster, is warranted.

## 1. Introduction

Mitochondria play an essential role during gestation, and their correct functioning is fundamental for optimal pregnancy development and resolution [1], protecting against offspring alterations [2] and maternal pathologies [3]. Mitochondria are multifunctional organelles involved in stem cell differentiation, programmed cell death, stress response [4], intracellular calcium concentration control, ATP production [5,6], and toxic waste regulation [4]. Mitochondria are also the leading producers of intracellular reactive oxygen species (ROS) [7]. The mitochondrial DNA copy number (mtDNAcn) is a highly sensitive biomarker for stress molecules from endogenous factors and environmental exposures. Environmental exposures and other oxidative stress-provoking factors have been associated with an altered mtDNAcn since duplication is the mechanism to eliminate damaged genetic material [8,9,10]. 

Children’s mitochondriomics could be modified by psychosocial stress, and its completion could enrich our knowledge in this science [11]. A recent study found that maternal lifetime trauma can modify the effect of prenatal exposure to particulate matter on placental mtDNAcn [12]. Experiencing a natural disaster during pregnancy can trigger acute stress that can affect the developing offspring [13,14]. Earthquakes have been previously linked to adverse perinatal outcomes, such as low birth weight, smaller head circumference, and early delivery [15,16,17]. However, to our knowledge, no studies have been conducted on the effect of prenatal exposure to acute stress resulting from this type of natural disaster on changes in the mtDNAcn in the offspring. 

Latin America is a region with high seismic activity. Chile, Ecuador, Haiti, Peru, Guatemala, and Mexico are among the countries that have experienced the strongest and most deadly earthquakes [18]. Mexico experienced a 7.8 Mw earthquake in 1957 [19], but from 1985 (8.1 Mw event) until 2017, higher-impact earthquakes did not occur in Mexico City, home to over 22 million people. On 19 September 2017, at 13:14 h, an intense (7.1 Mw) earthquake hit the city, causing severe damage to buildings, loss of lives, and generating overall chaos. This study aimed to compare the mtDNAcn in the umbilical cord blood of infants born before, those in gestation during, and those conceived up to a year after the earthquake. 

## 2. Materials and Methods

### 2.1. Study Population

The data came from the ongoing OBESO (Biochemical and Epigenetic Origins of Overweight and Obesity) perinatal cohort in Mexico City, which studies obesity and maternal metabolic profile as predictors of fetal body composition, obesity, and neurodevelopment during infancy. Women were invited to participate at the National Institute of Perinatology (INPer); they were recruited in their first trimester (11.0–13.6 weeks determined by ultrasound) and followed throughout pregnancy and the child’s first years. Inclusion criteria included the following: being ≥18 years old, no previous comorbidities (i.e., diabetes mellitus, hypertension, thyroid alterations, cardiopathies, autoimmune, renal, or hepatic diseases, or HIV), a pregestational BMI ≥ 18.5, no current treatment that can affect carbohydrate or lipid metabolism (i.e., insulin, metformin, steroids), and a fetus without structural congenital malformations. The first participant was recruited in January 2017, and the first child was born in July of the same year. The project was approved by the institutional review board of the INPer (3300-11402-01-575-17), and all women signed informed consent before participating.

Our study included 83 mother–infant pairs: *n* = 22 infants born before 19 September 2017 (non-exposed group), *n* = 24 newborns from women who were pregnant when the earthquake happened and were born between 18 October 2017 and June 2018 (exposed during pregnancy), and *n* = 37 newborns conceived after the earthquake and born between July 2018 and June 2019 (exposed before pregnancy). 

### 2.2. Mitochondrial DNA Copy Number Quantification

Umbilical cord blood was collected at birth using vacutainer tubes with EDTA and processed within 2 h of the collection. Blood was centrifuged at 1500 rpm for 10 min at room temperature, and the buffy coat was separated, washed, and stored at −80 °C until DNA extraction. DNA was isolated using the wizard DNA extraction kit (PROMEGA Madison, MDN, WI, USA), following the manufacturer’s instructions. Quantification of mtDNAcn was performed in triplicate by quantitative real-time polymerase chain reaction (qRT-PCR), using 12.5 ng of DNA and QuantiTect SYBR^®^ Green (QIAGEN, Hilden, DE), reactions were run in a StepOne™ Real-Time PCR System (Applied Biosystems™, WLM, MA, USA) with previously described PCR conditions [20,21]. The mtDNAcn was determined through the quantification of the mitochondrial gene (mt) mt-ND1, as described by Janssen [20], and normalized with the unique copy nuclear gene (S) of Beta Globin (HBG-β), as reported by Hou [21], calculating the mt/S ratio. A DNA pool from the cohort with a 10–0.001 ng/uL range (serial dilutions 1:10) was used to construct a standard curve to determine the concentration of the mt and S genes and calculate the mt/S ratio. The variation coefficient was 3% for S and 4% for mt. 

### 2.3. Covariates

Detailed information was obtained on maternal pregestational body mass index (BMI), socio-demographic characteristics (maternal age and socioeconomic status), and gender, weight, length, and head circumference of newborns. Information on pregnancy comorbidities (i.e., gestational diabetes and preeclampsia) developed during pregnancy was also collected. Pre-existing stress was evaluated using the perceived Stress Scale-4 (PSS-4) collected during the 1st and 3rd trimester visits [22].

### 2.4. Statistical Analysis

We analyzed the normality of our data using the Kolmogorov–Smirnov test and compared the maternal characteristics and comorbidities (preeclampsia, gestational diabetes), as well as mtDNAcn, of non-exposed newborns, those exposed during pregnancy, and those conceived after the earthquake, compared by one-way ANOVA for continuous variables, and the chi-square test for non-parametric data. A linear regression model was used to analyze the association between exposure groups and mtDNAcn adjusted by maternal age, BMI, pre-existing stress, and pregnancy comorbidities. Additionally, based on previous evidence linking perinatal susceptibility to maternal age [23], the association between maternal age and newborn mtDNAcn stratified by earthquake exposure timing was explored. Data were expressed as the mean ± standard deviation (SD). All analyses used Rstudio software version 1.4 with R version 4.0. Values of *p* < 0.05 were considered statistically significant.

## 3. Results

The results of the demographic, socioeconomic, and clinical characteristics of the mother–newborn dyads are shown in Table 1. Overall, no differences were found in maternal age (*p* = 0.817) and maternal pregestational BMI (*p* = 0.8) among the study groups. A total of 42 gave birth to a girl, and 41 gave birth to a boy. Almost half of the women reported middle socioeconomic status, which was not significantly different between the study groups (*p* = 0.73). Perceived stress in the first trimester was different compared to the third trimester of pregnancy, and no significant difference among the study groups was found in maternal comorbidities, such as preeclampsia and gestational diabetes. Figure 1 illustrates how mtDNAcn was lowest in the non-exposed newborns (mean = 0.740 ± 0.161), higher in those whose mothers were pregnant during the earthquake (mean = 0.899 ± 0.156), and highest among those conceived after the earthquake (mean = 0.995 ± 0.169). 

The results of the progressively adjusted regression model show that compared to non-exposed newborns, newborns whose mothers were pregnant during the earthquake had a significant increase in mtDNAcn (*p*-value < 0.001), and this increase was greater for those whose mothers were pregnant after the earthquake (*p*-value < 0.001), without losing statistical significance (Table 2). Demographic characteristics, perceived stress in the first and third trimesters, and comorbidities were not associated with mtDNAcn in either group.

The results of the interaction analysis of predictor variables show that increasing maternal age had a positive association with the mtDNAcn of newborns (β = 0.022, *p* = 0.007) in the group exposed during pregnancy, whereas in the non-exposed group and group exposed after the earthquake, there was no association (β = −0.002, *p* = 0.91; β = −0.012, *p* = 0.17) (Figure 2).

## 4. Discussion

In this study, we found that infants born to women who were pregnant during the 2017 Mexico City earthquake had a higher mtDNAcn than those born before the earthquake. Interestingly, we observed a stronger mtDNAcn increase in infants conceived after the earthquake than in those born before this event, which could imply that acute stress during pregnancy can have both short- and long-term consequences. 

Understanding the biological implication of mtDNAcn increase or decrease has emerged as a relatively new area of research. We found no reference for what can be considered a normal mtDNAcn; studies have usually used their population as a reference, and the parameters reported are only valid for that population. Furthermore, mtDNAcn is tissue specific and dependent on the stage of life [24]. During pregnancy, mitochondrial metabolism and function adapt to different adverse conditions such as intrauterine growth restriction to protect the fetus and reach full-term gestation [2]. There is evidence that an increase in placental mtDNAcn is associated with a reduced intellectual capacity in children [25] since mitochondria are fundamental for intelligence [26]. The response of mtDNAcn could be associated with experiencing trauma and stress in utero and can be independent of the concurrent maternal response [12]. mtDNAcn has previously been used as a marker of abuse and other life adversities in preschoolers [27]; moreover, this response can be maintained for several years and might be associated with depression and anxiety, among other psychological problems [28]. Later in life, studies have also found that an increase in plasma mtDNAcn is associated with post-traumatic stress [29], depression [30], severe depression, and anxiety [28,31], as well as suicide disorder [32]. Adverse experiences can condition long-term mitochondrial dysfunction; for instance, an increased mitochondrial response has been found in war veterans [28] and orphans or subjects experiencing domestic violence or sexual abuse [28,33].

Our results are also in line with studies about chemical environmental exposures during pregnancy that have found an association with altered mtDNAcn in the offspring; however, results are inconclusive on whether the copy number increases or decreases. A study showed a positive association between third-trimester manganese exposure and cord blood mtDNAcn [34]; other studies have found negative associations with gestational exposure to particulate matter, thallium, carbon monoxide, arsenic, and mercury [20,35,36,37,38,39]. Although evidence of co-exposure to chemical and psychosocial stress exposures is limited, a study showed a reduced mtDNAcn associated with co-exposure to PM_2.5_ and maternal lifetime trauma [12]. 

Intriguingly, infants conceived after the earthquake had an even higher mtDNAcn than those born to women pregnant during the time of the earthquake. We hypothesized that this could be associated with post-traumatic stress [40] and the fact that we have continued to have strong earthquakes in Mexico City. We had information on perceived stress during the first and the third trimesters of pregnancy and found no direct association with mtDNAcn. Perceived stress is very different from the acute stress that women most likely experienced due to the earthquake. Soon after the earthquake, we implemented a post-traumatic stress scale; however, this was not applied systematically and was not included in this study. Another hypothesis is that the acute stress experienced during the earthquake remains a memory in maternal mitochondria and is inherited by the newborn [41]. A study of prenatal exposure to stress from the Quebec ice storm of 1998 and offspring DNA methylation showed that 13 years later, epigenetic changes persisted, which are known as the DNA methylation signature [42]. Future studies in our cohort will be able to answer these questions. To confirm the increase in the newborn mtDNAcn of women exposed prior to their pregnancy, maternal measurements would be needed at the time of the earthquake or at a date close to it, to compare and follow up the effect at that time point, and to explore whether this effect was preserved until the delivery of their child. The OBESO cohort follow-up begins from the first trimester of pregnancy; therefore, we do not have samples from the mothers before their pregnancies. This is a limitation of our work.

The 2017 earthquake affected thousands of inhabitants of Mexico City, with over 200 deaths and 400 injured persons, plus extended physical damage to the buildings and infrastructure of the city. It was the strongest earthquake felt in the city since 1985, which happened on the same calendar date and had thousands of casualties. Beyond psychosocial stress, earthquakes can physically affect individuals due to the environmental changes they generate [43]. In the pre-seismic activity, an electromagnetic change occurs in the Earth’s crust that alters animal behavior. Radon emissions can last for an average of 2.8 days [44], ionizing the air, peri-oxidizing water, and generating an environment of oxidative stress [45]. We considered that these facts could influence the biological mechanism behind the mitochondrial response; however, this was beyond the scope of our research. Although we cannot account for the study participants’ location at the time of the earthquake, an inclusion criterion for the cohort was residing in Mexico City. The earthquake happened on a working day, during working hours.

An additional finding was the difference in the association between maternal age and mtDNAcn depending on the exposure. For those women that experienced the earthquake while pregnant, there was a statistically significant increase in mtDNAcn in their newborns as maternal age increased. mtDNAcn depends on different factors such as tissue type, sex, and age [24]. We hypothesized that the association seen with age could be explained if older women’s perception of post-traumatic stress and danger is enhanced [46,47], and this could condition future events; it has been shown that post-traumatic stress susceptibility increases with age after an earthquake [48].

Most environmental health studies focus on chemical exposures; the results of our study add to the existing research and highlight the importance of including stress, and in this case, acute stress, as a relevant prenatal exposure. As is true of many chemicals, with stress, there can be overlapping or similar mechanisms that are affected and therefore similar health outcomes. For example, previous studies have shown that co-exposure to stress and lead is associated with the functioning of the hypothalamic–pituitary–adrenal axis [49]. Our results support the need for increased awareness in medical practitioners and screening for acute stress exposure during pregnancy, as well as a closer follow-up of the offspring. Previous studies have identified mtDNAcn as a marker of the fetal physiological state that could be predictive of health risks in adult life. In adults, altered mtDNAcn has been associated with cognitive loss [50], cardiovascular outcomes [51], and increased susceptibility to infectious diseases [52].

To our knowledge, this is the first study linking mtDNAcn to acute stress derived from an earthquake. We are unaware of other studies of mtDNAcn alterations as a result of prenatal exposure to stress derived from a natural disaster. In this sense, this study is a pioneer in evaluating a biological response to this type of event. We are aware of our limited sample size; however, our cohort was uniquely positioned to carry out this study since we were collecting umbilical cord blood as part of our research protocol. Unfortunately, the cohort started only a few months prior to the earthquake, and after it, recruitment slowed down dramatically because some areas of the city were severely damaged. We are unaware of other birth cohorts that collected umbilical cord blood in Mexico City before and after the earthquake, and which also analyze mtDNAcn. Since previous studies have shown associations between stress and epigenetic changes, such as Project Ice Storm (REF), we could speculate that if the sample size were increased, the associations would be stronger, narrowing the confidence intervals. Therefore, although this is a small sample, this study provides evidence for future work, not only in Mexico (a country with high seismic activity) but also in others with similar stressful natural disasters. 

Lastly, another limitation of our study was the analysis of mtDNAcn only in umbilical cord blood. In order to gain a more complete picture of the mother–infant biological response, and to answer whether this response is generalized or tissue specific, analysis of mtDNAcn in different tissues, such as maternal blood or placental tissue, would be desirable. 

OBESO is an ongoing cohort; we look forward to continuing the follow-up of these children, including their neurodevelopment and the long-term permanence of the mitochondrial response. 

## 5. Conclusions

The findings of this study point at mtDNAcn as a potential biological marker of acute stress and suggest that experiencing an earthquake during pregnancy or before gestation can have programing effects in the unborn child. Future studies should clarify if an altered mtDNAcn could reflect susceptibility to diseases immediately or later in life.

## Figures and Tables

**Figure 1 ijerph-18-11771-f001:**
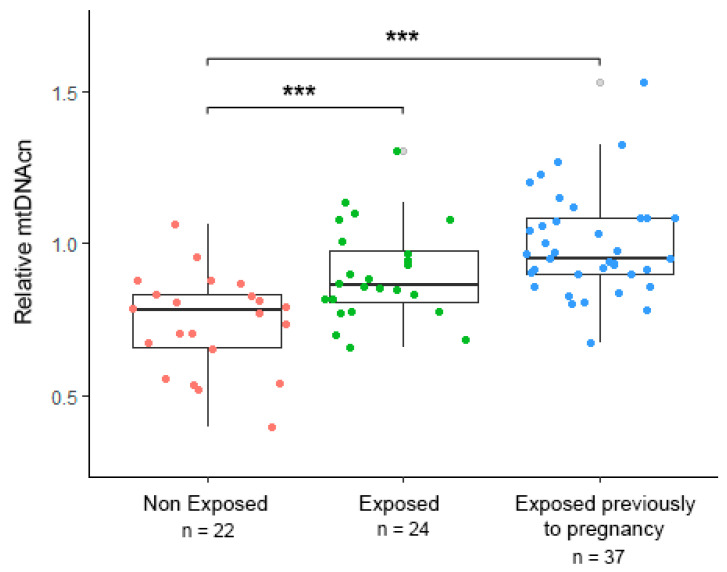
Umbilical cord blood mitochondrial DNA copy number (mtDNAcn) in infants born before the earthquake (non-exposed, *n* = 22, red circles), who were in gestation when the earthquake happened (exposed during pregnancy, *n* = 24, green circles), and who were conceived after the earthquake (exposed after pregnancy, *n* = 37, blue circles). Statistical differences were *p* < 0.05 (*** = *p* < 0.001).

**Figure 2 ijerph-18-11771-f002:**
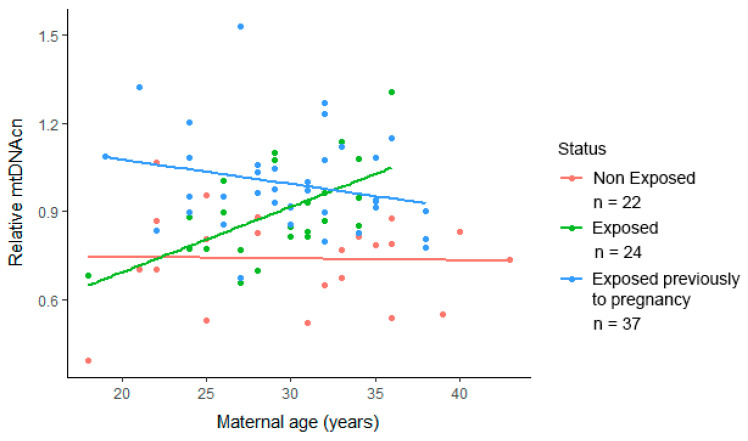
Maternal age as an effect modifier in the adjusted association between earthquake exposure during pregnancy and mtDNAcn, *p* < 0.05. Data are presented in a scatter plot with a linear regression by study group.

**Table 1 ijerph-18-11771-t001:** Characteristics of pregnant women and newborns exposed and non-exposed to the earthquake.

Characteristic	Non-Exposed (*n* = 22)	Exposed during Pregnancy (*n* = 24)	Exposed after Pregnancy (*n* = 37)	*p*-Value
Maternal age (years)	30 ± 7	29 ± 4	30 ± 5	0.817
Maternal pregestational body mass index (kg/m^2^)	26.35 ± 4.03	26.86 ± 4.36	27.3 ± 6.53	0.8
Low socioeconomic status	5 (22.7)	4 (16.6)	6 (16.2)	0.819
Middle socioeconomic status	13 (59.1)	12 (50)	19 (51.3)	0.376
High socioeconomic status	4 (18.2)	8 (33.3)	12 (32.4)	0.135
1st trimester pre-existing stress	9.86 ± 4.58	10.8 ± 3.66	9.03 ± 3.17	0.205
3rd trimester pre-existing stress	8.55 ± 3.19	8.68 ± 3.33	8.21 ± 3.03	0.879
Preeclampsia	3 (13.6)	4 (16.6)	4 (10.8)	0.913
Gestational diabetes	1 (4.5)	2 (8.3)	2 (5.4)	0.819
Newborns				
Gestational age at birth (weeks)	37.6 ± 2.1	38.5 ± 1.2	38.6 ± 1.4	0.156
Weight (g)	2755 ± 643	2938 ± 340	2837 ± 333	0.37
Length (cm)	45.4 ± 2.8	46.7 ± 1.4	47.3 ± 1.8	0.06
Head circumference (cm)	33 ± 2.1	33.3 ± 1.3	33.4 ± 1.1	0.529
Sex				
Female	8 (36.4)	12 (50)	22 (59.5)	0.07
Male	14 (63.6)	12 (50)	15 (40.5)	0.843

Data expressed as mean ± standard deviation, or frequency and percentage.

**Table 2 ijerph-18-11771-t002:** Association between earthquake exposure and newborn umbilical cord mtDNAcn, progressively adjusted for covariates. The infants born before the earthquake were the reference group.

Statistical Model	Earthquake during Gestation (*n* = 24) (Regression Coefficient)	95% Confidence Interval	Gestation after the Earthquake (*n* = 37) (β (SE))	95% Confidence Interval	AIC
mtDNAcn~Status	0.159	(0.072, 0.241)	0.255	(0.176, 0.344)	−60.34
mtDNAcn~Status + Sex	0.174	(0.086, 0.253)	0.275	(0.196, 0.363)	−62.73
mtDNAcn~Status + Sex + BMI	0.174	(0.083, 0.252)	0.274	(0.194, 0.362)	−60.75
mtDNAcn~Status + Sex + BMI + Age	0.175	(0.083, 0.253)	0.276	(0.194, 0.363)	−58.83
mtDNAcn~Status + Sex + BMI + Age + CoMo	0.175	(0.087, 0.265)	0.277	(0.190, 0.362)	−56.31
mtDNAcn~Status + Sex + BMI + Age + CoMo + SES	0.170	(0.07, 0.265)	0.270	(0.174, 0.346)	−56.84
mtDNAcn~Status + Sex + BMI + Age + CoMo + SES + Stress 1T	0.168	(0.079, 0.276)	0.260	(0.178, 0.359)	−53.83
mtDNAcn~Status + Sex + BMI + Age + CoMo + SES + Stress 3T	0.186	(0.93, 0.296)	0.266	(0.181, 0.367)	−48.04

All *p*-values were < 0.001; AIC: Akaike information criterion, Sex: newborn sex, BMI: prenatal maternal body mass index, CoMo: maternal comorbidities (pregestational diabetes or preeclampsia), SES: socioeconomic status, Stress 1T (first trimester pre-existing stress), Stress 3T (third trimester pre-existing stress).

## Data Availability

The datasets generated and/or analyzed during the present study are not publicly available but are available from the corresponding author on reasonable request.

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
