# Peer review of "Mitochondrial DNA Copy Number Adaptation as a Biological Response Derived from an Earthquake at Intrauterine Stage"

_ijerph, 2021, doi:10.3390/ijerph182211771_

Round 1

Reviewer 1 Report

This is a single center prospective cohort study done in mexico city to determine the effect of earthquakes ( stress) on Mitochondrial DNA copy number (mtDNAcn) in Umbilical cord blood of infants born before during and after the event. This is an interesting study.

General comments:

  1. The cohort sample size for each group is rather small, do the authors believe that the statistics may be different when conducted with a large enough sample size?
  2. Do the authors think that this difference can be seen with any environmental stress and is there evidence that other natural disasters can cause similar findings of mtDNA alterations
  3. Why do the authors think that infants conceived much after the said event had even higher rate of mtDNAcn alterations vs those that were actually pregnant during the time of the earthquake? By the virtue of continuing stress how does one differentiate this from ongoing stressors in life and is there anyway to decipher if the women that conceived after the earthquake went through additional environmental stressors?
  4. From the discussions the authors show that multiple factors can lead to mtDNA changes- what is the clinical relevance of this particular finding then?
  5. Has there been any follow up on these infants to see if there is normalization at the molecular level after a given period of time when comparing these cohorts?

Author Response

We thank the reviewer for the time invested in our work. We provide a reply to each of your comments (in bold)  and indicate where in the manuscript we have made the according changes.

Reviewer 1: 

This is a single center prospective cohort study done in mexico city to determine the effect of earthquakes ( stress) on Mitochondrial DNA copy number (mtDNAcn) in Umbilical cord blood of infants born before during and after the event. This is an interesting study.

General comments:

  1. The cohort sample size for each group is rather small, do the authors believe that the statistics may be different when conducted with a large enough sample size?

We agree with the reviewers point, in order to increase the sample size we included women who got pregnant after the earthquake and our results did not change. Previous studies have shown associations between stress and epigenetic changes, like DNA methylation in Project Ice Storm. Our results are in line with these findings, therefore we could speculate that if the sample size would be increased, the associations would be stronger, narrowing the confidence intervals. Our cohort was uniquely positioned to carry out this study, since we were collecting umbilical cord blood as part of our research protocol. Unfortunately, the cohort had started only a few months prior to the earthquake and after the earthquake recruitment slowed down dramatically (since some areas of the city were severely damaged),  hence the small sample size. We are unaware of other birth cohorts that were collecting umbilical cord blood in Mexico City prior and after the earthquake, and who are analyzing mtDNAcn. Therefore, although this is a small sample, this study provides evidence for future work, not only in Mexico (a country with a high seismic activity) but to others with similar stressful natural disasters. We have added the following text in the discussion:

“We are aware of our limited sample size, however our cohort was uniquely positioned to carry out this study, since we were collecting umbilical cord blood as part of our research protocol. Unfortunately, the cohort had started only a few months prior to the earthquake and after it, recruitment slowed down dramatically (since some areas of the city were severely damaged). We are unaware of other birth cohorts that were collecting umbilical cord blood in Mexico City prior and after the earthquake, and who were also analyzing mtDNAcn. Since previous studies have shown associations between stress and epigenetic changes, like Project Ice Storm (REF), we could speculate that if the sample size would be increased, the associations would be stronger, narrowing the confidence intervals. Therefore, although this is a small sample, this study provides evidence for future work, not only in Mexico (a country with a high seismic activity) but to others with similar stressful natural disasters”.   

  1. Do the authors think that this difference can be seen with any environmental stress and is there evidence that other natural disasters can cause similar findings of mtDNA alterations

There is evidence of the association between prenatal exposure to chemicals and mtDNAcn alterations, however, we are unaware of other studies on stress derived from a natural disaster. In this sense, this study is a pioneer in evaluating a biological response to this type of event.  We have added the following text to the discussion:

“We are unaware of other studies of mtDNAcn alterations as a result of prenatal exposure to stress derived from a natural disaster. In this sense, this study is a pioneer in evaluating a biological response to this type of event”.

  1. Why do the authors think that infants conceived much after the said event had even higher rate of mtDNAcn alterations vs those that were actually pregnant during the time of the earthquake? By the virtue of continuing stress how does one differentiate this from ongoing stressors in life and is there anyway to decipher if the women that conceived after the earthquake went through additional environmental stressors?

The reviewer raises an interesting question. We were actually intrigued by this finding, and we hypothesize that this could be associated with post-traumatic stress [1], and the fact that we have continued to have strong earthquakes in Mexico City, however, we do not have the means to corroborate this in our study. We began assessing PTSD through a questionnaire after the earthquake, but unfortunately we don’t have enough women with results to include in the analysis. Another hypothesis is that the acute stress experienced during the earthquake remains as a memory in maternal mitochondria and is inherited by the newborn [2] . We have included the following text in the discussion:    

“Intriguingly, infants conceived after the earthquake had an even higher mtDNAcn than those born to women pregnant during the time of the earthquake, we hypothesize that this could be associated with post-traumatic stress (REF), and the fact that we have continued to have strong earthquakes in Mexico City….Another hypothesis is that the acute stress experienced during the earthquake remains as a memory in maternal mitochondria and is inherited by the newborn (REF)”.

To answer the reviewers second point, we had the results of a perceived stress questionnaire in the 1st and 3rd trimester of pregnancy, but this didn’t change our results. We could not apply the PTSD questionnaires systematically and could not account for this particular stress later. We are certain that shortly after the strong earthquake there were repeated weaker earthquakes, so women were likely exposed to these. When there is an earthquake of 5 or more Richter points an alarm is set off in Mexico City, this can trigger the same stress reaction since we are uncertain of the intensity or type of earthquake until the event is over.  

  1. From the discussions the authors show that multiple factors can lead to mtDNA changes- what is the clinical relevance of this particular finding then?

Most environmental health studies focus on chemical exposures, the results of our study add to the existing research and highlight the importance of including stress, and in this case acute stress as a relevant prenatal exposure. As is true of many chemicals, with stress, there can be overlapping or similar mechanisms that are affected and therefore similar health outcomes. For example, previous studies have shown that the co-exposure of stress and lead are associated with the functioning of the hypothalamic-pituitary-adrenal axis[3]. Our results could be translated to an increased awareness in medical practitioners and the screening for acute stress exposure during pregnancy, as well as a closer follow-up of the offspring. Previous studies have identified mtDNAcn as a marker of fetal physiological state that could be predictive of health risks in adult life. In adults, altered mtDNAcn has been associated with cognitive loss [4], cardiovascular outcomes [5] and increased susceptibility to infectious diseases [6].

We have added the following text to the discussion:

“Most environmental health studies focus on chemical exposures, the results of our study add to the existing research and highlight the importance of including stress, and in this case acute stress as a relevant prenatal exposure. As is true of many chemicals, with stress, there can be overlapping or similar mechanisms that are affected and therefore similar health outcomes. For example, previous studies have shown that the co-exposure of stress and lead are associated with the functioning of the hypothalamic-pituitary-adrenal axis (REF). Our results could be translated to an increased awareness in medical practitioners and the screening for acute stress exposure during pregnancy, as well as a closer follow-up of the offspring. Previous studies have identified mtDNAcn as a marker of fetal physiological state that could be predictive of health risks in adult life. In adults, altered mtDNAcn has been associated with cognitive loss (REF), cardiovascular outcomes (REF) and increased susceptibility to infectious diseases (REF)”.

  1. Bersani, F.S.; Morley, C.; Lindqvist, D.; Epel, E.S.; Picard, M.; Yehuda, R.; Flory, J.; Bierer, L.M.; Makotkine, I.; Abu-Amara, D.; et al. Mitochondrial DNA Copy Number Is Reduced in Male Combat Veterans with PTSD. Progress in Neuro-Psychopharmacology and Biological Psychiatry 2016, 64, 10–17, doi:10.1016/j.pnpbp.2015.06.012.
  2. Zhang, Q.; Wang, Z.; Zhang, W.; Wen, Q.; Li, X.; Zhou, J.; Wu, X.; Guo, Y.; Liu, Y.; Wei, C.; et al. The Memory of Neuronal Mitochondrial Stress Is Inherited Transgenerationally via Elevated Mitochondrial DNA Levels. Nat Cell Biol 2021, 23, 870–880, doi:10.1038/s41556-021-00724-8.
  3. Cory-Slechta, D.A.; Virgolini, M.B.; Rossi-George, A.; Thiruchelvam, M.; Lisek, R.; Weston, D. Lifetime Consequences of Combined Maternal Lead and Stress. Basic & Clinical Pharmacology & Toxicology 2008, 102, 218–227, doi:10.1111/j.1742-7843.2007.00189.x.
  4. Dolcini, J.; Kioumourtzoglou, M.-A.; Cayir, A.; Sanchez-Guerra, M.; Brennan, K.J.; Dereix, A.E.; Coull, B.A.; Spiro, A.; Vokonas, P.; Schwartz, J.; et al. Age and Mitochondrial DNA Copy Number Influence the Association between Outdoor Temperature and Cognitive Function. Environ Epidemiol 2020, 4, e0108, doi:10.1097/EE9.0000000000000108.
  5. Castellani, C.A.; Longchamps, R.J.; Sumpter, J.A.; Newcomb, C.E.; Lane, J.A.; Grove, M.L.; Bressler, J.; Brody, J.A.; Floyd, J.S.; Bartz, T.M.; et al. Mitochondrial DNA Copy Number Can Influence Mortality and Cardiovascular Disease via Methylation of Nuclear DNA CpGs. Genome Med 2020, 12, 84, doi:10.1186/s13073-020-00778-7.
  6. Fukunaga, H. Mitochondrial DNA Copy Number and Developmental Origins of Health and Disease (DOHaD). International Journal of Molecular Sciences 2021, 22, 6634, doi:10.3390/ijms22126634.

  1. Has there been any follow up on these infants to see if there is normalization at the molecular level after a given period of time when comparing these cohorts?

After the earthquake (Sept 2017), the recruitment and follow-up of the OBESO study was paused for a few months and then slowly picked up. Then, since March 2020, due to sanitary measures and security restrictions at the National Institute of Perinatolgy, it has been impossible for us to take samples from the infants belonging to the cohort. We have been unable to measure postpartum children’s mtDNAcn in to compare it with the measurement at birth. However, we have archived blood samples and we plan to analyze the samples in the near future. Then, we will be able to report the pediatric follow-up and the long-term permanence of the mitochondrial response.

In the last sentence of the discussion we have added:

As OBESO is an ongoing cohort, we will continue follow-up of these children, including their neurodevelopment and the long-term permanence of the mitochondrial response.

Reviewer 2 Report

This is a very interesting small study on mitochondrial DNA copy number in cord blood collected at birth in study groups their mothers who were exposed during and before pregnancy compared to non-exposed mothers. Linear regression showed a statistically significant increase of mitochondrial DNA copy number in newborn cord blood in mothers exposed during and a more pronounced increase in mothers exposed before the current pregnancy (during a prior pregnancy). There are a few major points that need to be addressed in the Discussion section. In addition, one sentence in the Abstract is unclear. Two ‘additions’ are required.

Major points:

  1. The sentence line 34-36 in the Abstract is unclear and not understandable. The presentation of the findings in the Result section are clear. Revise the sentence in the Abstract.
  2. Table 2, Figure 1 and 2: Sample sizes are required to be shown in the column of the exposure groups and in the legend of the figures.
  3. The Discussion should clarify that maternal blood would be required to determine long-term effects related to prior pregnancies of the mother. And it would be preferable in future investigations to also determine mitochondrial DNA copy number in maternal blood and the placenta to gain better understanding of processes related to the number of mitochondria.

Minor points:

  1. Line 118: Remove leading blank.
  2. Line 204: Italicize ‘in utero’.

Author Response

We thank the reviewer for the time invested in our work. We provide a reply to each of your comments (in bold)  and indicate where in the manuscript we have made the according changes.

Reviewer  2: 

This is a very interesting small study on mitochondrial DNA copy number in cord blood collected at birth in study groups their mothers who were exposed during and before pregnancy compared to non-exposed mothers. Linear regression showed a statistically significant increase of mitochondrial DNA copy number in newborn cord blood in mothers exposed during and a more pronounced increase in mothers exposed before the current pregnancy (during a prior pregnancy). 

We are pleased to know the reviewer found our study interesting, thank you.

There are a few major points that need to be addressed in the Discussion section. In addition, one sentence in the Abstract is unclear. Two ‘additions’ are required.

Major points:

  1. The sentence line 34-36 in the Abstract is unclear and not understandable. The presentation of the findings in the Result section are clear. Revise the sentence in the Abstract.

We thank the reviewer for this observation, we have modified the abstract and the results now read as follows:

Compared to non-exposed newborns (mean±SD mtDNAcn: 0.740 ± 0.161), exposed and post-earthquake newborns (mtDNAcn: 0.899 ± 0.156 and 0.995± 0.169 respectively) had increased mtDNAcn, p=0.001.

A few lines before we have clarified that: “...24 infants whose mothers were pregnant at the time of the  earthquake (exposed), and 37 who were conceived after the earthquake (post-earthquake)

  1. Table 2, Figure 1 and 2: Sample sizes are required to be shown in the column of the exposure groups and in the legend of the figures.

The corrected figures have been modified in the manuscript. 

  1. The Discussion should clarify that maternal blood would be required to determine long-term effects related to prior pregnancies of the mother. And it would be preferable in future investigations to also determine mitochondrial DNA copy number in maternal blood and the placenta to gain better understanding of processes related to the number of mitochondria.

This is an excellent observation, we agree that in able to ensure that the increase in newborn mtDNAcn of women exposed prior to their pregnancy, maternal measurements would be needed at the time of the earthquake or at a date close to it, to compare and follow-up the effect at that time point, and if this effect was preserved until the delivery of their child. The OBESO cohort follow-up begins from the first trimester of pregnancy, therefore we do not have samples from the mother before her pregnancy.  This is a limitation in our work. We have now included  which we justify with the background of the long-term effect of mtDNAcn in response to PTSD experienced in war veterans. We have included the following text in 2 places in the discussion:

“In able to ensure that the increase in newborn mtDNAcn of women exposed prior to their pregnancy, maternal measurements would be needed at the time of the earthquake or at a date close to it, to compare and follow-up the effect at that time point, and if this effect was preserved until the delivery of their child. The OBESO cohort follow-up begins from the first trimester of pregnancy, therefore we do not have samples from the mother before her pregnancy.  This is a limitation in our work”.

“Intriguingly, infants conceived after the earthquake had an even higher mtDNAcn than those born to women pregnant during the time of the earthquake, we hypothesize that this could be associated with post-traumatic stress (REF), and the fact that we have continued to have strong earthquakes in Mexico City”.

In response to the reviewer’s second comment, this is an excellent suggestion for future work. OBESO has archived samples of maternal blood and placental tissue, we will definitely take the reviewers suggestion and aim at securing the means to analyze these samples. An important point is that at this moment, we are unable to know for certain if we have the matching samples for the newborns included in this study, but we will take this suggestion in consideration for the near future, follow-up study, thank you. 

We have added the following text in the discussion:

Lastly, another limitation of our study was the analysis of mtDNAcn only in umbilical cord blood. In order to have a more complete picture of the mother-infant biological response, as well as to answer whether this response is generalized or tissue specific, analysis of mtDNAcn in different tissues, such as maternal blood or placental tissue would be desirable.